# Changes in rapid plasma reagin titers in patients with syphilis before and after treatment: A retrospective cohort study in an HIV/AIDS referral hospital in Tokyo

**Kazuhiko Ikeuchi**[1,2], **Kazuaki Fukushima**[1]*, **Masaru Tanaka**[1], **Keishiro Yajima**[1], **Makoto Saito**[2], **Akifumi Imamura**[1]

1 Department of Infectious Diseases, Tokyo Metropolitan Cancer and Infectious Diseases Center Komagome Hospital, Bunkyo-ku, Tokyo, Japan, 2 Division of Infectious Diseases, Advanced Clinical Research Center, Institute of Medical Science, The University of Tokyo, Minato-ku, Tokyo, Japan

* fukushima-ngi@umin.ac.jp

## Abstract

### Introduction

Although the rapid plasma reagin (RPR) test is used to determine treatment efficacy for syphilis, animal studies show that it decreases gradually after an initial increase even without treatment. Pre-treatment changes in RPR titer in humans and its relationship with post-treatment changes in RPR titer are not well known.

### Methods

We retrospectively analyzed the clinical records of syphilitic patients who underwent automated RPR (Mediace) testing twice before treatment (i.e., at diagnosis and treatment initiation) within 1–3 months at an HIV/AIDS referral hospital in Japan between 2006 and 2018. The RPR values were expressed as the ratio to the value at treatment initiation. The mean monthly relative change in the RPR after treatment was calculated on the log2 scale for each patient and analyzed by multivariable linear regression.

### Results

Sixty-eight patients were identified. The median age was 45 (interquartile range [IQR], 38–50), 98.5% (67/68) were men, and 97.1% (66/68) had HIV. The median RPR titer ratio at treatment initiation/diagnosis was 0.87 (IQR, 0.48–1.30). The RPR titer decreased more than twofold in 26.5% (18/68) and more than fourfold in 10.3% (7/68) before treatment. In the multivariable analysis, higher age (predicted monthly RPR relative change on the log2 scale 0.23/10 years [95% confidence interval [CI], 0.090–0.37]), history of syphilis (0.36 [95% CI, 0.07–0.65]), and a lower ratio of RPR at treatment initiation/diagnosis (−0.52/every 10-fold increase [95% CI, −0.81 to −0.22]) were associated with a slower RPR decrease after treatment.

**Data Availability Statement:** All relevant data are within the paper and its Supporting Information files.

**Funding:** The authors received no specific funding for this work.

**Competing interests:** The authors have declared that no competing interests exist.

## Conclusions

In a mostly HIV patient population, RPR titer can show more than four-fold spontaneous increase or decrease within 1–3 months. Pre-treatment spontaneous decrease of RPR titer was associated with a slower decrease in post-treatment RPR titer.

## Introduction

Syphilis is a sexually transmitted infection that is caused by the spirochete *Treponema pallidum* subsp. *pallidum*. Typically, *T. pallidum* causes primary syphilis at the invasion site and disseminates throughout the body, causing secondary syphilis and its associated symptoms, such as skin rashes [1]. *T. pallidum* invades the central nervous system (CNS) in 25%–60% of early syphilis, but 95% are asymptomatic [1]. Although *T. pallidum* that invades the CNS is eliminated in most patients with intramuscular or oral antibiotics [2,3], patients who fail to eliminate *T. pallidum* from CNS are at risk of progressing to symptomatic neurosyphilis [1], which requires intravenous treatment. Therefore, it is important to treat promptly with effective antibiotics and monitor the treatment efficacy.

Nontreponemal tests (e.g., rapid plasma reagin [RPR] test) detect antibodies against the cardiolipin-cholesterol-lecithin antigen. Nontreponemal tests are highly sensitive, cheap, and decrease in response to treatment and are used as a marker of treatment efficacy [4–6]. The current guidelines recommend that patients with primary and secondary syphilis should undergo nontreponemal tests at 6 and 12 months and, if the titer does not decrease fourfold, clinicians should consider cerebrospinal fluid (CSF) tests [7].

However, some researchers and clinicians have reported that the nontreponemal titer could change without treatment. In a rabbit model of syphilis infection, the natural course of the nontreponemal titer shows a bell-shaped curve with a longer right tail; the titer increases shortly after inoculation, then decreases gradually, and can become negative even without treatment [8]. We hypothesized that if nontreponemal titer increases and decreases spontaneously in humans, then pre-treatment changes in nontreponemal titer might affect post-treatment response.

Since even early latent syphilis is transmittable to other persons and can progress to symptomatic syphilis [9], immediate treatment should be started after diagnosis. Therefore, there are only a few data about the natural course of the nontreponemal titer before treatment, and it is unethical to conduct such prospective clinical studies. However, in the real world, patients may sometimes be unable to receive treatment on the day of diagnosis for some reasons [10]. For example, people with human immunodeficiency virus (HIV) (PWH) could be incidentally diagnosed with syphilis by routine screening for sexually transmitted infections, and unfortunately, treatment may be delayed to the next visit to the hospital particularly when the patients were asymptomatic. Using the routinely collected clinical data of syphilitic patients who had a gap between diagnosis and treatment, we conducted a retrospective cohort study to elucidate the pre-treatment changes in nontreponemal titer and of its influence on the post-treatment nontreponemal response.

## Methods

### Setting and patients

We retrospectively reviewed the clinical records of patients diagnosed with syphilis between January 2006 and December 2018 at Tokyo Metropolitan Cancer and Infectious Disease

Center Komagome Hospital, which is an HIV/acquired immune deficiency syndrome (AIDS) referral hospital in Tokyo, Japan. The records of the patients with syphilis whose RPR titer was measured twice before treatment (i.e., at diagnosis and at treatment initiation) were extracted for this study. Some of these patients were included in our earlier study [11]. An episode of syphilis was defined as an elevation of the RPR titer ($\geq$8.0 R.U. [RPR unit]) and a positive *T. pallidum* latex agglutination test result (Sekisui Medical, Mediace RPR, and TPLA). If the baseline RPR was positive (i.e., having a history of syphilis), a more than fourfold increase in RPR titer was defined as a new infection.

Patients with confirmed neurosyphilis by CNS tests and/or ocular syphilis were excluded. The Japanese medical insurance system allows a maximum of 3 months to prescribe medication and, therefore, the interval between the routine outpatient visits is usually 1–3 months. Patients were excluded if their interval between the time of diagnosis and treatment initiation were outside this stipulated period (i.e., <15 days and >105 days). For patients with multiple episodes, only the most recent episode was included in the analysis.

## Data collection

Information on the following variables were extracted from the clinical records: age, sex, history of syphilis, clinical stage of syphilis, RPR titer at diagnosis, RPR titer at treatment initiation, TPLA titer at treatment initiation, antibiotics, serological cure, HIV infection, CD4 counts, and HIV viral load. The RPR titer was assessed until it decreased by at least fourfold from the value recorded at treatment initiation. The RPR titer was followed until December 2019. Serological cure was defined as a fourfold decrease in the RPR titer within 12 and 24 months in early symptomatic syphilis and in latent syphilis, respectively.

Primary and secondary syphilis were classified as early symptomatic syphilis. Early latent syphilis was defined as asymptomatic syphilis acquired within 12 months. Early symptomatic syphilis and early latent syphilis were classified as early syphilis, and late latent syphilis and latent syphilis of unknown duration were classified as late syphilis.

The automated RPR test, which is commonly used in Japan and Korea [12,13], is a quantitative method for evaluating the agglutination reaction between anti-lipid antibodies and latex. The coefficient of variation (CV) of the conventional manual RPR tests is reported to be 25%–40%, whereas the CV of the automated RPR titer is reported to be 10%–13% around the cut-off RPR titer (1.0 R.U), and only <5% at higher levels [14]. Considering the CV of the automated RPR test, a ratio of RPR at treatment initiation/diagnosis of <80%, 80–120%, and >120% was defined as indicating a decreased-, stable-, and increased-RPR group, respectively.

This study was approved by the Institutional Committee on Research Ethics of our hospital (approval number: 2990). This study was conducted in an opt-out method on our hospital website to obtain informed consent.

## Statistical analysis

First, the pre-treatment change in the RPR titer was illustrated using the ratio of RPR titer at treatment initiation/diagnosis. Second, we assessed whether the pre-treatment change in the RPR titer (i.e., the ratio of RPR measurements at treatment initiation/diagnosis) could affect the slope of the post-treatment relative change in the RPR titer. The ratio (i.e., relative change) of the RPR titer at post-treatment/treatment initiation was used for the analysis. The averaged monthly change in the RPR ratio (i.e., speed of the decrease in RPR) after treatment was calculated on the log2 scale for each patient by using the slope of linear approximation (i.e., log-linear was assumed). The mean of the monthly change in the RPR ratio was compared between the groups on the log2 scale using the Student's *t*-test and antilog was used for describing the

mean and 95% Confidence Intervals (CI). Univariable and multivariable linear regression analyses were conducted to assess the variables that affect the monthly change in RPR titer ratio. Variable selection was conducted by backward elimination using the *P*-values obtained with the Wald test. Statistical significance was defined as a 2-sided *P*-value <0.05. For highly correlated variables, one value was chosen based on clinical importance. Data analysis was performed using R version 4.0.2 and Stata MP 16.1 (StataCorp, TX, USA).

## Results

Between January 2006 and December 2018, 395 patients experienced 522 syphilis infection (Fig 1). After excluding 25 neurosyphilis and/or ocular syphilis, 371 patients (93.9%, 371/395) had 497 syphilis infections (95.2%, 497/522). Among them, 94 patients (25.3%, 94/371) experienced 111 syphilis infections (22.3%, 111/497) in which RPR titer was tested twice before treatment: at diagnosis and at treatment initiation. Twenty-six patients were excluded because the gap between diagnosis and treatment initiation was outside the defined time period (<15 days, n = 1; >105 days, n = 25). A total of 68 patients were tested for RPR titer twice before treatment in 1–3 months. They experienced 79 such infection episodes and their last episodes were analyzed.

The interval between diagnosis and treatment initiation was 1, 2, and 3 months in 9 (13.2%), 39 (57.4%), and 20 patients (29.4%), respectively. The median age was 45 years (interquartile range [IQR], 38–50 years), and 98.5% (67/68) were men. Among them, 66 men were PWH. Thirty-eight patients (55.9%, 38/68) had a history of syphilis. Most patients were treated

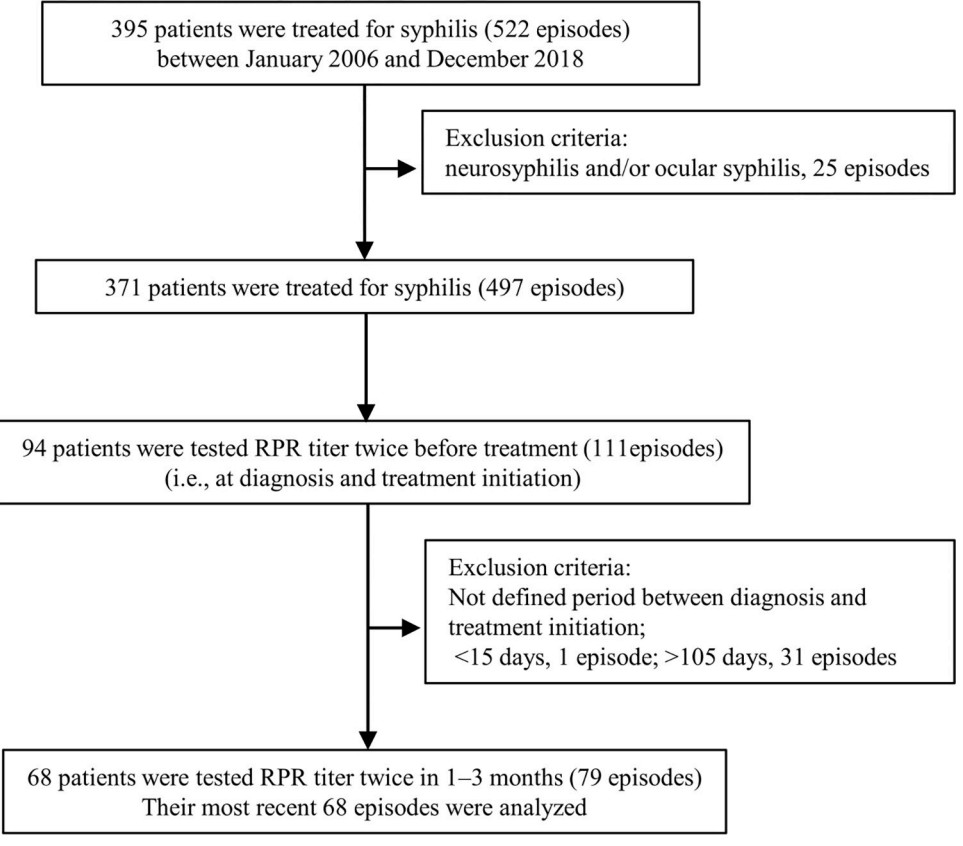

**Fig 1. Flow chart of the study procedures.** Abbreviations: RPR, Rapid plasma regain.

with oral or intravenous regimen (amoxicillin, n = 45; oral benzylpenicillin, n = 22; intravenous penicillin G, n = 1), because intramuscular benzathine penicillin G had not been authorized in the study period in Japan. The median RPR titer was 163 R.U. (IQR, 56–328 R.U.) at diagnosis and 132 R.U. (IQR, 62–252 R.U.) at treatment initiation. The median RPR titer ratio at treatment initiation/diagnosis was 0.87 (range, 0.11–44.25; IQR, 0.48–1.30). The RPR titer increased more than twofold in 17.6% (12/68) and fourfold in 11.8% (8/68), whereas it decreased more than twofold in 26.5% (18/68) and fourfold in 10.3% (7/68) of patients before treatment. After treatment, serological cure was achieved in all patients who were followed RPR titer for defined duration (58/58). RPR titers were not followed until serological cure in the rest.

Based on the change in the RPR titer before treatment, the patients were categorized into three groups: increased-RPR group (n = 20), stable-RPR group (n = 18), and decreased-RPR group (n = 30). The temporal changes in the RPR titer from before and after treatment for each patient are shown in Fig 2, and the characteristics of each group are shown in Table 1. The mean monthly change in the ratio of the RPR titer after treatment was 0.47-fold/month (95% confidence interval [CI], 0.35–0.63/month) on linear scale in the increased-RPR group (the reference group), 0.59-fold/month (95% CI, 0.46–0.76/month) in the stable-RPR group (*P* = 0.23), and 0.71-fold/month (95% CI, 0.65–0.78/month) in the decreased-RPR group (*P* = 0.002).

The results of the linear regression are shown in Table 2. In the univariable analysis, higher age, HIV-RNA >50 copies/mL, late-stage syphilis, history of syphilis, and a lower ratio of RPR titer at treatment initiation/diagnosis were associated with larger slopes of post-treatment RPR, which means a slower post-treatment decrease (i.e., the downslope becoming closer to zero). In the multivariable analysis, higher age (predicted mean monthly change in the RPR ratio on the log2 scale 0.23/10 years [95% CI, 0.090–0.37], *P* = 0.002), history of syphilis (0.36 [95% CI, 0.07–0.65], *P* = 0.02) and a lower ratio of RPR at treatment initiation/diagnosis (−0.52/every 10-fold increase [95% CI, −0.81 to −0. 22], *P* = 0.001) were associated with a slower post-treatment decrease in RPR. Other variables were not associated with the post-treatment relative changes in RPR titer.

## Discussion

We have shown that the RPR titer could change dramatically within 1–3 months even without treatment. Surprisingly, the RPR titer decreased spontaneously by fourfold in 10% of patients in our cohort. Furthermore, the pre-treatment decreases in the RPR titer, in addition to the higher age and a history of syphilis were significantly associated with a slower post-treatment decline in the RPR titer.

Our data showed that the RPR titer could decrease more than fourfold before treatment, which means that the treatment success criteria could be achieved without treatment in some patients. In a rabbit model, non-treponemal titers increased after infection and then spontaneously decreased more than fourfold within 6 months even without treatment [8]. Even after the RPR titer became negative, however, viable *T. pallidum* was detected in the lymph nodes of rabbits. Therefore, even patients with low RPR titers need to receive antibiotic treatment if they have not received appropriate treatment.

Recently, Pandey K, *et al.* reported that RPR changed more than fourfold in 14.8% of syphilis patients within 14 days before treatment [10], despite the difference that most patients were non-PWH and the RPR card test (Becton Dickinson) was used in their study. In their study, fewer patients showed a fourfold decrease in pre-treatment RPR titer than those with an increase (increase, 12.3%; decrease, 2.5%), while they were similar in our study (increase,

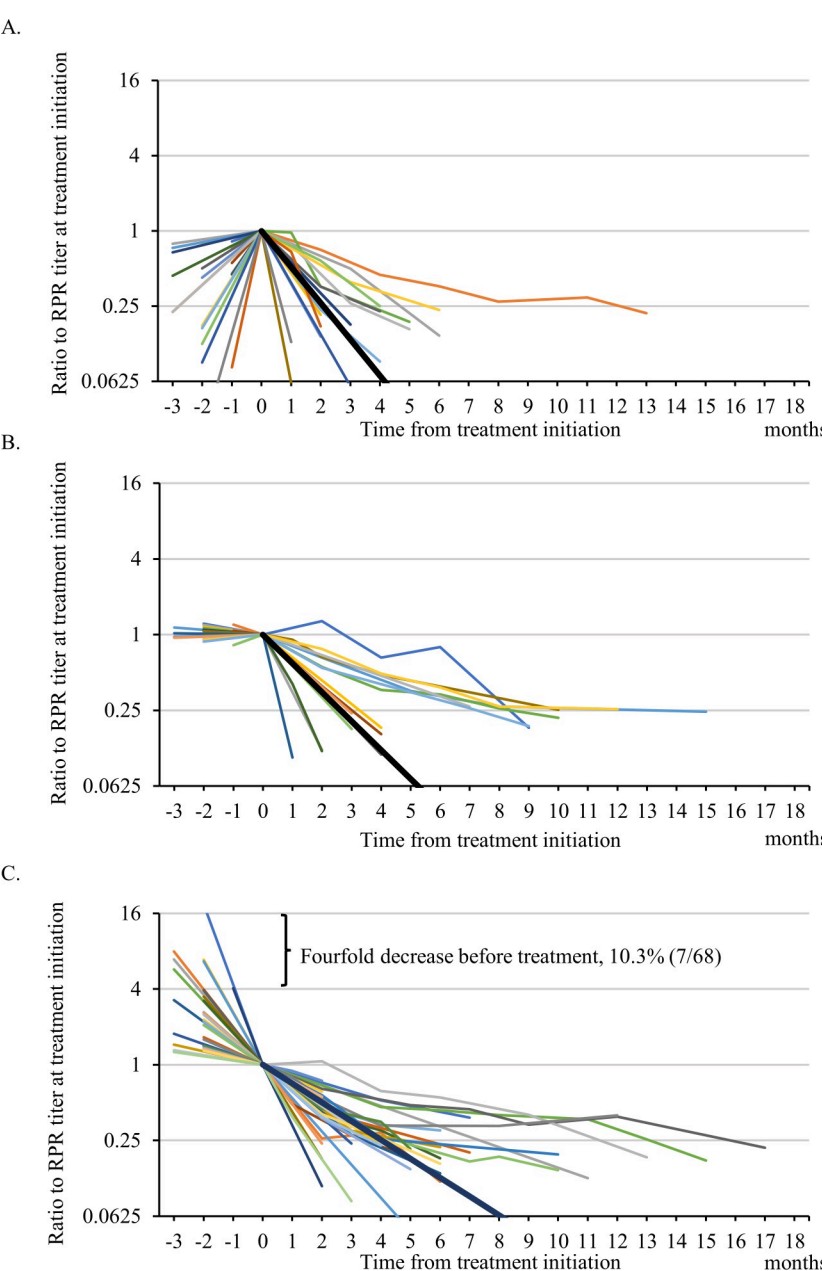

**Fig 2. Temporal changes in the RPR titer from before to after treatment.** The time course of the RPR titer from before to after treatment in each group is shown (2A, increased-RPR group; 2B, stable-RPR group; 2C, decreased-RPR group). Each line represents the time course of the RPR titer for each patient. The black line represents the average post-treatment RPR change in each group. Abbreviations: RPR, rapid plasma regain.

11.8%; decrease, 10.3%). In the rabbit model, the spontaneous RPR decrease is much slower than the initial increase. The longer duration of the pre-treatment period in our study can be one reason for the higher proportion of patients with decreased titer observed in our study.

The strength of our study is that we evaluated the association between pre- and post-treatment RPR change, in addition to the description of pre-treatment changes in RPR titer. Of note, the lower ratio of the RPR titer at treatment initiation/diagnosis, rather than a lower RPR titer at treatment initiation, was associated with a slower post-treatment RPR decrease. This

**Table 1. Clinical characteristics of patients in the increased-, stable-, and decreased-RPR groups.**

| | Increased-RPR group | | Stable-RPR group | | Decreased-RPR group | |
|---|---|---|---|---|---|---|
| **Characteristics** | **n = 20** | | **n = 18** | | **n = 30** | |
| Age (years) | 46 | (41–50) | 44 | (40–51) | 45 | (37–51) |
| Male | 20 | (100%) | 17 | (94.4%) | 30 | (100%) |
| HIV infection | 20 | (100%) | 16 | (88.9%) | 30 | (100%) |
| CD4 counts (/μL) | 508 | (349–675) | 496 | (316–609) [a] | 482 | (370–605) |
| HIV-RNA <50 copies/mL | 17 | (85.0%) | 14 | (87.5%) [a] | 26 | (86.7%) |
| Stage of syphilis | | | | | | |
| Early symptomatic | 5 | (25.0%) | 4 | (22.2%) | 5 | (16.7%) |
| Early latent | 14 | (70.0%) | 9 | (50.0%) | 20 | (66.7%) |
| Late latent | 1 | (5.0%) | 1 | (5.6%) | 0 | (0%) |
| Late with unknown duration | 0 | (0%) | 4 | (22.2%) | 5 | (16.7%) |
| History of syphilis | 12 | (60.0%) | 6 | (33.3%) | 20 | (66.7%) |
| RPR titer at diagnosis (R.U.) | 64 | (19–261) | 110 | (76–262) | 229 | (92–359) |
| RPR titer at treatment initiation (R.U.) | 212 | (126–377) | 118 | (73–248) | 103 | (30–183) |
| TPLA at treatment initiation (T.U.) | 12,045 | (4,187–24,935) | 7,985 | (4,756–16,615) | 7,619 | (3,231–16,470) |
| Treatment | | | | | | |
| Oral amoxicillin | 14 | (70.0%) | 11 | (61.1%) | 20 | (66.7%) |
| Oral benzylpenicillin | 5 | (25.0%) | 7 | (38.9%) | 10 | (33.3%) |
| Intravenous benzylpenicillin G | 1 | (5.0%) | 0 | (0%) | 0 | (0%) |
| Serological cure | 20 | (100%) | 15 | (100%) [b] | 23 | (100%)[b] |
| Mean monthly change in the ratio of the RPR titer after treatment [c] | 0.47 | (0.35–0.63) | 0.59 | (0.46–0.76) | 0.71 | (0.65–0.78) |

Data are presented as median (interquartile range) or number (percentage) unless otherwise specified.

[a] Only people living with HIV (n = 66) were analyzed.

[b] RPR titers were not followed until serological cure in 10 patients (RPR-stable group, n = 3; RPR-decreased group, n = 7).

[c] Mean (95% confidence interval) were calculated on log2 scale and are expressed in antilog.

**Abbreviations:** RPR, rapid plasma reagin; R.U., RPR unit; TPLA, *Treponema pallidum* latex-agglutination; T.U., titer unit.

**Table 2. Association between the slope of the ratio of the post-treatment RPR titer compared to the baseline titer and other characteristics using linear regression analysis.**

| | Univariable | | | Multivariable | | |
|---|---|---|---|---|---|---|
| | **Coefficient** | | ***P*-value** | **Coefficient** | | ***P*-value** |
| Age (per 10 years) | 0.23 | (0.067–0.39) | 0.006 | 0.23 | (0.090–0.37) | 0.002 |
| CD4 counts (per 100μL) | 0.066 | (−0.017 to 0.15) | 0.12 | | | |
| HIV-RNA <50 copies/mL | 0.67 | (0.19–1.16) | 0.008 | | | |
| Late syphilis* | 0.37 | (−0.09 to 0.82) | 0.12 | | | |
| History of syphilis | 0.45 | (0.12–0.78) | 0.008 | 0.36 | (0.07–0.65) | 0.02 |
| RPR titer at diagnosis (per 10R.U.) | 0.0060 | (−0.00036 to 0.012) | 0.06 | | | |
| RPR titer at treatment initiation (per 10R.U.) | 0.0020 | (−0.0078 to 0.012) | 0.69 | | | |
| Ratio of RPR titer at treatment initiation/diagnosis (per every 10-fold increase) | −0.57 | (−0.90 to −0.25) | 0.001 | −0.52 | (−0.81 to −0.22) | 0.001 |
| TPLA titer (per 100T.U.) | 0.00054 | (−0.00072 to 0.0018) | 0.40 | | | |
| Amoxicillin therapy | −0.27 | (−0.63 to 0.08) | 0.13 | | | |

Coefficients are shown in log2 scale. **Abbreviations:** RPR, rapid plasma reagin; R.U., RPR unit; TPLA, *Treponema pallidum* latex-agglutination; T.U., titer unit.

*compared with early syphilis.

result is highly understandable if the natural course of RPR titer in humans is similar to that of rabbits (i.e., a bell-shaped curve): when the treatment is initiated during the increasing or apex of the bell-shaped curve (i.e., increased-RPR group and some patients in the stable-RPR group), the rate of post-treatment RPR decrease is expected to be faster, because the RPR titer decreases spontaneously even without treatment; however, when the treatment is initiated during the decreasing or right-tail of the curve (i.e., stable-RPR group and decreased-RPR group), the rate of the post-treatment RPR decrease is expected to be slower. However, it is impossible to ascertain the pre-treatment RPR change in most cases who are treated promptly. Thus, even if the RPR titer, stage, and age at treatment initiation were the same, the speed of the post-treatment RPR decrease could be completely different depending on the timing from the infection. Generally, a slow post-treatment RPR decrease is used as an indicator of treatment failure, mainly due to neurosyphilis, but there is little evidence [9]. Future studies are needed to determine whether clinicians should evaluate neurosyphilis in all patients with a slow decrease in post-treatment RPR.

It has been reported that the slow RPR decrease or serological cure after treatment is associated with various factors, such as higher age, a more advanced stage of syphilis, prior history of syphilis, and lower baseline RPR titers [15]. Among them, higher age was associated with a slower RPR decrease in our multivariable model, too. Based on our hypothesis, it is reasonable that the advanced stage of syphilis, in which RPR titers have already decreased spontaneously, is associated with slow RPR decrease after treatment, but we did not find a statistically significant association between the advanced stage and slow RPR decrease in the present study. It could be attributed to the limited number of late syphilis, comprising only two patients with definite late-stage latent syphilis, while nine patients had syphilis with unknown duration. Similar to the present study, prior history of syphilis was associated with a slower RPR response in some studies, indicating potential differences in post-treatment RPR change between initial infection and reinfection [15–17]. A lower baseline RPR titer is a well-known risk factor for slower RPR decrease after treatment [15,17], but pre-treatment RPR decrease was more significant than baseline RPR titer itself in the multivariable analysis in our study. A lower baseline RPR titer might be a result of a pre-treatment RPR decrease.

HIV infection also known to affect the baseline RPR titer and its response after treatment. For example, HIV is a major risk factor for serofast (i.e., more than four-fold decrease in RPR titer but still testing positive), resulting in higher baseline RPR titers in repeated infection in PWH [17–19]. Biological false-positives are also common in PWH due to the production of antiphospholipid antibodies caused by dysregulated B cell activation [19]. Most patients were PWH in our study and more than half had a history of syphilis, so these backgrounds might affect the baseline RPR titer and the speed of RPR decrease. Immunocompromised status, such as CD4 <350/μL and untreated HIV also reported as risk of serofast [20,21]. Although HIV-RNA <50 copies/mL was associated with a slow RPR decrease in our univariable analysis, it is counterintuitive that HIV-uncontrolled patients had a better treatment course for syphilis [21]. This apparent association between HIV-RNA and RPR change was probably confounded by age and the history of syphilis in our cohort.

It is not ideal that many patients were not treated on the date of diagnosis in our hospital (15.1% of total infections, 79/522). This percentage was not so different from the previous study (15.6%, 766/4903) [10]. When PWH are diagnosed with latent syphilis by screening blood tests, they usually leave the hospital before the results become available. Although PWH do not miss their syphilis treatment as they are scheduled for their next visit, to prevent progression of the disease and the potential spread of infection [9], clinicians should contact them for prompt treatment even if they are asymptomatic. In addition, we should reevaluate the RPR titer, because it might be different from diagnosis.

There are several limitations of our study. First, we used an automated RPR test, which is not a global standard. The automated RPR test "Mediace" used in this study correlated strongly with

the conventional manual RPR card test [13]. Mediace automated RPR has been reported to be less sensitive when titers are low [14,22], but the titer in this study was high enough in most cases. Compared to the conventional manual RPR tests, the seroconversion rate after 12 months of treatment in Mediace RPR tests was almost the same, but the Mediace RPR titer was reported to decrease often faster [13,22]. Future studies are needed to determine whether our results are reproducible when the conventional manual method is used. Second, because most patients were PWH in the present study, serological response could have been different from non-PWH as discussed above. While the guidelines state that there is no need to change the criteria for treatment success for PWH [9], they point out that nontreponemal titer may decrease slower. Third, this was a retrospective study that followed the RPR titer for only 1–3 months. To reveal the natural course of RPR, it is required to follow patients with syphilis without treatment for a long time prospectively, but this is ethically unacceptable. Fourth, the treatment of non-neurosyphilis in Japan is different from that in many other countries. Intramuscular benzathine penicillin G, the gold standard for the treatment of syphilis, was not authorized in Japan until 2021. Therefore, the patients treated by intramuscular benzathine penicillin G were not included in this study. In Japan, oral benzylpenicillin was previously used as an alternative to intramuscular benzathine penicillin G in the 2000s, and was gradually replaced by oral amoxicillin in Japan because of the low bioavailability of oral benzylpenicillin. In previous studies, oral amoxicillin showed approximately 95% efficacy for non-neurosyphilis, and approximately 90% efficacy for even late-stage latent syphilis [11,23]. The slope of the post-treatment RPR titer in our study might be different from that after treatment with intramuscular penicillin G. Fifth, we did not perform CSF tests for patients with slower post-treatment RPR response to rule out the asymptomatic neurosyphilis. However, in general, a higher RPR titer is a risk factor for neurosyphilis [24,25], so it is unreasonable to hypothesize that a patient with a decreasing RPR titer is more likely to have neurosyphilis. Whether the slow RPR decrease is useful in predicting neurosyphilis requires further study. Sixth, there is a possibility that patients received antibiotics before our treatment. However, because antibiotics are not available over the counter without a prescription in Japan, self-treatment is almost impossible. Furthermore, approximately 80% of participants were asymptomatic syphilis patients; therefore, there was no reason for them to take antibiotics for syphilis by themselves. Although there is a possibility that they received antibiotics for other reasons in other hospitals, it is difficult to explain the high percentage of patients with decreased pre-treatment RPR titers.

## Conclusions

Approximately 10% of patients with syphilis achieved "serological cure", based on the RPR titer, before treatment. When we cannot start treatment on the day of diagnosis for some reasons, we should reevaluate the RPR titer and, even if RPR titer decreases, should treat syphilis when patients have not been treated before. Higher age, a history of syphilis, and a lower rate of RPR titer at treatment initiation and diagnosis were associated with a slower post-treatment RPR decrease. Therefore, even if patients have the same characteristics and the same RPR titer at treatment initiation, the rate of RPR decrease after treatment could be quite different. Because our results depend on the automated RPR tests from PWH treated with oral regimen, further studies are desired for the generalizability of our findings.

## Supporting information

**S1 File. * months from treatment initiation.** RPR ratio compared with RPR at treatment initiation is shown.
(XLSX)

## Acknowledgments

We would like to thank Editage (www.editage.com) for English language editing.

## Author Contributions

**Conceptualization:** Kazuhiko Ikeuchi, Kazuaki Fukushima.

**Data curation:** Kazuhiko Ikeuchi, Kazuaki Fukushima, Masaru Tanaka, Keishiro Yajima, Akifumi Imamura.

**Formal analysis:** Kazuhiko Ikeuchi.

**Investigation:** Kazuhiko Ikeuchi.

**Methodology:** Kazuhiko Ikeuchi, Makoto Saito.

**Project administration:** Kazuhiko Ikeuchi, Kazuaki Fukushima.

**Supervision:** Kazuaki Fukushima, Makoto Saito, Akifumi Imamura.

**Visualization:** Kazuhiko Ikeuchi.

**Writing – original draft:** Kazuhiko Ikeuchi.

**Writing – review & editing:** Kazuhiko Ikeuchi, Masaru Tanaka, Keishiro Yajima, Makoto Saito, Akifumi Imamura.

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
