## [Decision Letter · Decision Letter 0]

12 Apr 2023

PONE-D-22-35264The natural course of nontreponemal antibody titers in patients with syphilis before treatment: A retrospective cohort studyPLOS ONE

Dear Dr. Fukushima,

Thank you for submitting your manuscript to PLOS ONE. After careful consideration, we feel that it has merit but does not fully meet PLOS ONE’s publication criteria as it currently stands. Therefore, we invite you to submit a revised version of the manuscript that addresses the points raised during the review process. Dear Authors, Your study was found interesting but I agree with the first referee that it has important and severe limitations. In particular, all the statements included in the text are based on the use of Mediace aRPR and Sekure RPR. Both methods require improvement before they can be used to diagnose syphilis or evaluate treatment efficacy in clinical practice. Second, some statements should be mitigated as they sound at least hazardous (in some cases, […], the elevation of nontreponemal titer was identified […] and treatment may be initiated at the next visit to the hospital after a few months). Third, the point that has raised the greatest concerns: all the post treatment statements are based on the routinary use of other antibiotics than benzathine penicillin G which is the first line therapy and gold standard for treatment of early syphilis according to current guidelines. This should be discussed and data about the use of standard benzathine penicillin G should be added Finally, the title and abstract should be modified.

We look forward to receiving your revised manuscript.

Kind regards,

Antonella Marangoni, Ph.D.

Academic Editor

PLOS ONE

Journal Requirements:

“No”

Additional Editor Comments:

Dear Authors,

Your study was found interesting but I agree with the first referee that it has important and severe limitations.

In particular, all the statements included in the text are based on the use of Mediace aRPR and Sekure RPR. Both methods require improvement before they can be used to diagnose syphilis or evaluate treatment efficacy in clinical practice.

Second, some statements should be mitigated as they sound at least hazardous (in some cases, […], the elevation of nontreponemal titer was identified […] and treatment may be initiated at the next visit to the hospital after a few months).

Third, the point that has raised the greatest concerns: all the post treatment statements are based on the routinary use of other antibiotics than benzathine penicillin G which is the first line therapy and gold standard for treatment of early syphilis according to current guidelines.

This should be discussed and data about the use of standard benzathine penicillin G should be added

Finally, the title and abstract should be modified

Reviewers' comments:

Reviewer's Responses to Questions

**Comments to the Author**

1. Is the manuscript technically sound, and do the data support the conclusions?

Reviewer #1: No

Reviewer #2: Yes

2. Has the statistical analysis been performed appropriately and rigorously? 

Reviewer #1: Yes

Reviewer #2: Yes

3. Have the authors made all data underlying the findings in their manuscript fully available?

Reviewer #1: Yes

Reviewer #2: Yes

4. Is the manuscript presented in an intelligible fashion and written in standard English?

Reviewer #1: Yes

Reviewer #2: Yes

5. Review Comments to the Author

Reviewer #1: Title of the article:

The natural course of nontreponemal antibody titers in patients with syphilis before treatment: A retrospective cohort study

Reference number:

PONE-D-22-35264

Comments:

Thank you for the opportunity to review your interesting manuscript. I enjoyed reading it as I consider the topic of this manuscript of great relevance. I have to say that the title might be misleading, and some statement should be mitigated throughout the text as serology in syphilis is a tricky issue (see Major Compulsory Revisions). The abstract should be carefully reorganized as the aims are not explicit and the results are not fully coherent with the title of the text (see Major Compulsory Revisions). Moreover, the conclusion paragraph of the manuscript is missing (see Major Compulsory Revisions). The main issue here is the use of Mediace automated RPR (aRPR) and Sekure rapid plasma reagin (RPR-S) (Sekisui Diagnostics) which accuracy and sensitivity require improvement before they can be used to diagnose syphilis and evaluate treatment efficacy in clinical practice (see Major Compulsory Revisions). A thorough focus in regards of the limitations of the tests used should be included to make this article suitable for publication.

- Major Compulsory Revisions

Line 2-3: Please, consider modifying the title of your manuscript. The title of the text is “The natural course of nontreponemal antibody titers in patients with syphilis before treatment: A retrospective cohort study”. However, 98.5% (67/68) are men and, moreover, 97.1% (66/68) are PLWH (people living with HIV). Additionally, a consistent part of the manuscript regards the serological response after treatment. It is known that PLWH are considered a special population in regards of their serological response [Ren M. AIDS. 2020].

Line 40-41: Please, consider explicating your aims and purposes in the abstract introduction. Moreover, a consistent part of your manuscript regards the serological response after treatment in a special population. Please, include this aspect in your aims part here.

Line 56-58: The abstract conclusion part might be misleading as the presented results might not be applicable to the general population. In your manuscript there is no evidence that more reliable markers of treatment efficacy are required in patients with syphilis. Please, try to mitigate your statement as 97.1% (66/68) of the sample considered are PLWH.

Line 74: Please, try to mitigate this sentence. Currently, various treponemal-specific immunoassays are increasingly being used for syphilis screening (reverse approach) including EIAs, CIAs, MBIAs, among others. These assays can be automated and are relatively nonspecific. For this reason, a nontreponemal test (eg, RPR) is used as a confirmatory test on initially reactive specimens [Park IU, Clin Infect Dis. 2020]

Line 89-93: Writing this sentence might be hazardous. As international guidelines suggest at least an annual screening for syphilis in PLWH, it is undoubtedly true that a non-treponemal titer elevation might occur unexpectedly. However, starting syphilis treatment “few months” after the prove of an infection might be considered at least hazardous as the patient is contagious and a danger for himself and the population. I would rather re-write this sentence pointing out your efforts in treating as soon as possible new cases of syphilis enlightening that, occasionally (due to the difficulty in contacting patients to schedule a treatment visit) few months might elapse between the diagnosis and treatment.

Line 104-106: The main issue here is the use of Mediace automated RPR (aRPR) and Sekure rapid plasma reagin (RPR-S) (Sekisui Diagnostics). These tests present the advantage of being automated, but Mediace aRPR has the disadvantage of a poor sensitivity in low titers [Leroy AG, Diagn Microbiol Infect Dis. 2022] and the sensitivity and accuracy of the RPR-S test requires improvement before it can be used to diagnose syphilis and evaluate treatment efficacy in clinical practice [Osbak K, J Med Microbiol. 2017]. In the discussion (line 195-200) the authors point out the pros of using aRPR. However, a thorough focus in regards of the limitations of the tests used must be included to make this article suitable for publication.

Line 180: Please, it is crucial to report the descriptive analysis of the treatment part. In other words, it is essential to explicit that none of the patient received intramuscular benzathine penicillin G (which is, as you stated in the discussion part, the gold standard for treatment of early syphilis).

Line 272-278: The conclusion part is missing. Please, insert this paragraph in your manuscript. Please, mitigate the statements in this paragraph in light of the limitation of the study (non-treponemal assays used, antimicrobial therapy administered, etc).

- Minor Essential Revisions

Line 70-72: It has been recently proven that polymerase chain reaction (PCR) tests might detect a T. pallidum infection even in individuals with a negative diagnostic serological test for syphilis [Junejo MH, Sex Transm Infect. 2022]. Perhaps it is not correct to state that nontreponemal tests are used for monitoring treatment efficacy “as a result” of a PCR low sensitivity especially thinking about early or primary syphilis. Please, try to re-organize these sentences pointing out the several pros of using non treponemal tests in clinical practice, according to the current guidelines.

Line 80-82: The popular study cited [Zhou P, Sex Transm Infect. 2012] has several limitations as the authors stated in their manuscript. I would rather enlighten in the text that the analysis cited is a case series in which multiple different antimicrobial regimens have been used to treat secondary syphilis.

Line 99: Please, clarify the methods used to rule out neurosyphilis. Please, explain in this paragraph how a diagnosis of “non-neurosyphilis” has been made.

Line 105: Please, make explicit the abbreviation and acronyms never used before in the manuscript.

Line 161-163: Among the 68 included, please, enlighten in the text how many patients had an history of syphilis (as you have already done in Table 1). This is crucial as serological trend in case of reinfection, especially among PLWH, might be anomalous [Marchese V, J Clin Med. 2022].

Line 169-171: Please, clarify this sentence.

Level of interest:

This is an interesting work as serological trend in syphilis infection is a tricky issue which requires further studies to better understand both the serological natural course and the serological response to treatment.

Reviewer #2: This interesting paper retrospectively evaluated the natural course of RPR titers in patients with syphilis before treatment, and also identified the association between changes in RPR titers before treatment and that after treatment. The data on natural course of RPR titers are limited since performing a prospective study is unethical. Through analyzing RPR testing results at diagnosis and at treatment initiation, the study found that RPR titers can spontaneously decrease before treatment and post-treatment decrease was slower in patients whose titers decreased before treatment. I have some minot comments as follows.

Minor comments:

Results:

1. Page 9, Line 164: The abbreviation "R.U." should be defined upon first appearance.

2. A recently published study (Pandey K et al. Clin Infect Dis. 2023;76(5):795-799.) showed 2.5% patients had ≥4-Fold decrease and 12.3% patients had ≥4-Fold increase in RPR titers before treatment. The differences between the 2 studies could be discussed in the Discussion section.

3. Page 10, Line 170: The authors claimed that "RPR titers were not followed until serological cure in 10 patients; however, based on the predicted change of the posttreatment RPR titer, it was expected that all patients would achieve serological cure." How do the authors perform the prediction?

Discussion:

4. Typo: page 11, line 194: "were" significantly associated with...

5. Page 11, Line 203: The included patients might still receive antibiotics with activity against T. pallidum from other hospitals/clinics for other reasons (e.g. pharyngitis, pneumonia, cellulitis...). If information on medication use from other hospitals/clinics are not available, the bias should be stated in the limitation.

6. Page 13: The association between posttreatment RPR decrease and history of syphilis was not discussed.

7. Page 14, Line 257: The reason of unavailability of IM BPG in Japan could be provided in short or with a cited reference.

8. Page 14, Line 264: The treatment efficacy of BPG enhanced with oral amoxicillin vs BPG alone was compared in a RCT conducted by Rolfs RT et al, which showed similar efficacy between the 2 treatment groups (83% vs 82%).

Tables:

9. Table 2: The coefficient and p values in multivariable analysis were different from that in the manuscript text.

10. Table 2: The reason that the variable "HIV-RNA <50 copies/mL" was not selected in the multivariable analysis should be stated.

6. PLOS authors have the option to publish the peer review history of their article (what does this mean?). If published, this will include your full peer review and any attached files.

Reviewer #1: No

Reviewer #2: No

---

## [Author Response · Author response to Decision Letter 0]

20 May 2023

Editor comments to the author

1. In particular, all the statements included in the text are based on the use of Mediace aRPR and Sekure RPR. Both methods require improvement before they can be used to diagnose syphilis or evaluate treatment efficacy in clinical practice.

Response: Thank you for your important advice. As you mentioned, automated RPR is not a global standard method for diagnosis and evaluation of treatment efficacy. We used Mediace RPR method in this study. According to the reviewers’ comments, we revised the title, abstract and discussions to clear the limitations of aRPR. Please see our reply to the seventh comment from reviewer 1.

2) Second, some statements should be mitigated as they sound at least hazardous (in some cases, […], the elevation of nontreponemal titer was identified […] and treatment may be initiated at the next visit to the hospital after a few months).

Response: Thank you for your comment. We deleted potentially hazardous description and mitigated this expression. Please see our reply to the sixth comment from reviewer 1.

3) Third, the point that has raised the greatest concerns: all the post treatment statements are based on the routinary use of other antibiotics than benzathine penicillin G which is the first line therapy and gold standard for treatment of early syphilis according to current guidelines. This should be discussed and data about the use of standard benzathine penicillin G should be added.

Response: Thank you for pointing out an important problem. In Japan, intramuscular penicillin G became authorized only in November 2021. Amoxicillin or doxycycline had been the first-line treatment for early syphilis recommended by Japanese guidelines. We don’t have much data of post-treatment RPR change after intramuscular penicillin G use in Japan. Please also see our reply to the eighth and nineth comments from reviewer 1 and the seventh and eighth comments from reviewer 2.

5) Finally, the title and abstract should be modified

Response: Thank you for your advice. We revised the title and abstract as follows: “Changes in rapid plasma reagin titers in patients with syphilis before and after treatment: a retrospective cohort study in an HIV/AIDS referral hospital in Tokyo.”

Reviewer 1

1. I have to say that the title might be misleading, and some statement should be mitigated throughout the text as serology in syphilis is a tricky issue (see Major Compulsory Revisions). The abstract should be carefully reorganized as the aims are not explicit and the results are not fully coherent with the title of the text (see Major Compulsory Revisions). Moreover, the conclusion paragraph of the manuscript is missing (see Major Compulsory Revisions). The main issue here is the use of Mediace automated RPR (aRPR) and Sekure rapid plasma reagin (RPR-S) (Sekisui Diagnostics) which accuracy and sensitivity require improvement before they can be used to diagnose syphilis and evaluate treatment efficacy in clinical practice (see Major Compulsory Revisions). A thorough focus in regards of the limitations of the tests used should be included to make this article suitable for publication.

Response: Thank you for your important and helpful comments. According to your comments, we carefully revised title, abstract and manuscripts as below.

- Major Compulsory Revisions

2. Line 2-3: Please, consider modifying the title of your manuscript. The title of the text is “The natural course of nontreponemal antibody titers in patients with syphilis before treatment: A retrospective cohort study”. However, 98.5% (67/68) are men and, moreover, 97.1% (66/68) are PLWH (people living with HIV). Additionally, a consistent part of the manuscript regards the serological response after treatment. It is known that PLWH are considered a special population in regards of their serological response [Ren M. AIDS. 2020].

Response: Thank you very much for your advice. We agree with your comment and changed the title to “Changes in rapid plasma reagin titers in patients with syphilis before and after treatment: a retrospective cohort study in an HIV/AIDS referral hospital in Tokyo.” 

3. Line 40-41: Please, consider explicating your aims and purposes in the abstract introduction. Moreover, a consistent part of your manuscript regards the serological response after treatment in a special population. Please, include this aspect in your aims part here.

Response: Thank you for your advice. We clarified our objective of this study as the association between pre- and post-treatment changes in automated RPR (title, Lines 38-41, 56-58, 80-81, 89-92 in the clear version).

4. Line 56-58: The abstract conclusion part might be misleading as the presented results might not be applicable to the general population. In your manuscript there is no evidence that more reliable markers of treatment efficacy are required in patients with syphilis. Please, try to mitigate your statement as 97.1% (66/68) of the sample considered are PLWH.

Response: Thank you for your important comment. We revised title and abstract accordingly to clarify that this study used the data collected at an HIV/AIDS referral hospital (title, Lines 42-44, 56-58 in the clear version). We also added this point to the Limitations (lines 254-264 in the clear version). 

5. Line 74: Please, try to mitigate this sentence. Currently, various treponemal-specific immunoassays are increasingly being used for syphilis screening (reverse approach) including EIAs, CIAs, MBIAs, among others. These assays can be automated and are relatively nonspecific. For this reason, a nontreponemal test (eg, RPR) is used as a confirmatory test on initially reactive specimens [Park IU, Clin Infect Dis. 2020]

Response: Thank you for your comment. The sentence is misleading and the specificity of non-treponemal and treponemal tests is not necessarily important in this context, so we removed this sentence (Line 70 in the clear version). 

6. Line 89-93: Writing this sentence might be hazardous. As international guidelines suggest at least an annual screening for syphilis in PLWH, it is undoubtedly true that a non-treponemal titer elevation might occur unexpectedly. However, starting syphilis treatment “few months” after the prove of an infection might be considered at least hazardous as the patient is contagious and a danger for himself and the population. I would rather re-write this sentence pointing out your efforts in treating as soon as possible new cases of syphilis enlightening that, occasionally (due to the difficulty in contacting patients to schedule a treatment visit) few months might elapse between the diagnosis and treatment.

Response: Thank you for your helpful advice. We emphasized that clinicians should ideally start treatment as soon as the diagnosis was made, in the introduction section (Lines 82-89 in the clear version). In addition, according to your advice, we added a paragraph that enlightens that treatment often delays in the real world and that treatment should be initiated as soon as possible (Lines 246-253 in the clear version).

7. Line 104-106: The main issue here is the use of Mediace automated RPR (aRPR) and Sekure rapid plasma reagin (RPR-S) (Sekisui Diagnostics). These tests present the advantage of being automated, but Mediace aRPR has the disadvantage of a poor sensitivity in low titers [Leroy AG, Diagn Microbiol Infect Dis. 2022] and the sensitivity and accuracy of the RPR-S test requires improvement before it can be used to diagnose syphilis and evaluate treatment efficacy in clinical practice [Osbak K, J Med Microbiol. 2017]. In the discussion (line 195-200) the authors point out the pros of using aRPR. However, a thorough focus in regards of the limitations of the tests used must be included to make this article suitable for publication.

Response: Thank you for your important comment. We modified the title and abstract to indicate that we used an automated RPR test. We agree that these automated RPR tests are controversial and not necessarily accepted globally. Therefore, we have moved the discussion of automated RPR tests to the limitation section from the section on the strength of this study (Title, Lines 43, 254-261 in the clear version).

8. Line 180: Please, it is crucial to report the descriptive analysis of the treatment part. In other words, it is essential to explicit that none of the patient received intramuscular benzathine penicillin G (which is, as you stated in the discussion part, the gold standard for treatment of early syphilis).

Response: Thank you for your advice. In Japan, unfortunately, intramuscular benzathine penicillin G became available only in November 2021, and oral amoxicillin or doxycycline was used for syphilis except neurosyphilis. This was the reason why none of the patients in our study were treated with intramuscular benzathine penicillin G. Although the lack of use of this global first-line treatment is a limitation of our study, we believe that this will not affect our findings on pre-treatment changes in RPR (Lines 267-276 in the clear version). We added this point to the results section (Lines 164-167 in the clear version).

9. Line 272-278: The conclusion part is missing. Please, insert this paragraph in your manuscript. Please, mitigate the statements in this paragraph in light of the limitation of the study (non-treponemal assays used, antimicrobial therapy administered, etc).

Response: Thank you for your important comment. We made the conclusion section, and mitigated the expression. We removed the sentence that the RPR test is unreliable for determining treatment efficacy. We also added the limitations to the conclusion section (Lines 289-298 in the clear version).

- Minor Essential Revisions

10. Line 70-72: It has been recently proven that polymerase chain reaction (PCR) tests might detect a T. pallidum infection even in individuals with a negative diagnostic serological test for syphilis [Junejo MH, Sex Transm Infect. 2022]. Perhaps it is not correct to state that nontreponemal tests are used for monitoring treatment efficacy “as a result” of a PCR low sensitivity especially thinking about early or primary syphilis. Please, try to re-organize these sentences pointing out the several pros of using non treponemal tests in clinical practice, according to the current guidelines.

Response: Thank you for your advice. We revised the explanation of nontreponemal tests. We removed the misleading description of PCR tests. We emphasized the strength of nontreponemal tests in this paragraph (Lines 70-72 in the clear version).

11. Line 80-82: The popular study cited [Zhou P, Sex Transm Infect. 2012] has several limitations as the authors stated in their manuscript. I would rather enlighten in the text that the analysis cited is a case series in which multiple different antimicrobial regimens have been used to treat secondary syphilis.

Response: Thank you for your comment. According to your advice, we chose not to emphasize the inaccuracy of RPR as a method of determining treatment efficacy. Therefore, we removed this reference (Line 80).

12. Line 99: Please, clarify the methods used to rule out neurosyphilis. Please, explain in this paragraph how a diagnosis of “non-neurosyphilis” has been made.

Response: Thank you for your important comment. As you pointed out, “non-neurosyphilis” is not a widely used term. We added the confirmed neurosyphilis and/or ocular syphilis as the exclusion criteria. We also revised the study flow chart (Lines 106, 154-155 in the clear version, Figure 1).

13. Line 105: Please, make explicit the abbreviation and acronyms never used before in the manuscript.

Response: Thank you for pointing it out. We added the abbreviation (Line 102 in the clear version, Table 1, Table 2)

14. Line 161-163: Among the 68 included, please, enlighten in the text how many patients had an history of syphilis (as you have already done in Table 1). This is crucial as serological trend in case of reinfection, especially among PLWH, might be anomalous [Marchese V, J Clin Med. 2022].

Response: Thank you for your comments. We added this point to the manuscript (Lines 163-164 in the clear version).

15. Line 169-171: Please, clarify this sentence.

Response: Thank you for your comment. The problems in this sentence were also pointed out by reviewer 2. We expected it from the log2 slope of RPR decrease, but it is not a usual way to predict the treatment success. Because it is misleading, we removed this sentence (Line 173 in the clear version).

Reviewer 2

Minor comments:

1. Page 9, Line 164: The abbreviation "R.U." should be defined upon first appearance.

Response: Thank you for your comment. “R.U.” is RPR unit, and “T.U.” is titer unit. We added the abbreviation (Line 102 in the clear version, Table 1, Table 2).

2. A recently published study (Pandey K et al. Clin Infect Dis. 2023;76(5):795-799.) showed 2.5% patients had ≥4-Fold decrease and 12.3% patients had ≥4-Fold increase in RPR titers before treatment. The differences between the 2 studies could be discussed in the Discussion section.

Response: Thank you for your advice. We have already read this recently published article after the initial submission of our article. Study design of this study is similar to ours, but we analyzed longer pre-treatment periods (1-3 months vs < 14 days). Another difference and the novelty of our study is that we analyzed the association with pre- and pos-treatment RPR change. Patient characteristics (low proportion of PWH) and RPR method (conventional manual method) were different, too. We added a new paragraph to explain these differences (Lines 206-213).

3. Page 10, Line 170: The authors claimed that "RPR titers were not followed until serological cure in 10 patients; however, based on the predicted change of the posttreatment RPR titer, it was expected that all patients would achieve serological cure." How do the authors perform the prediction?

Response: Thank you for your pointing out. We have removed this sentence as we have received the same comment from reviewer 1. Please see also our reply to the 15th comment from reviewer 1.

4. Typo: page 11, line 194: "were" significantly associated with...

Response: Thank you for your comment. We revised this sentence (Line 198 in the clear version).

5. Page 11, Line 203: The included patients might still receive antibiotics with activity against T. pallidum from other hospitals/clinics for other reasons (e.g. pharyngitis, pneumonia, cellulitis...). If information on medication use from other hospitals/clinics are not available, the bias should be stated in the limitation.

Response: Thank you for your comment. The lack of information on the use of antibiotic use outside of our hospital is an important limitation, even though antibiotics are not available over the counter in Japan and it is not very likely to receive oral antibiotics for such a long time for minor infections. We added this point to the limitation (Lines 281-287 in the clear version).

6. Page 13: The association between posttreatment RPR decrease and history of syphilis was not discussed.

Response: Thank you for your comments. We added the discussion (Lines 240-242) as follows: “Prior history of syphilis, associated with a slower RPR response in the present study, was associated in some studies but not in others.”

7. Page 14, Line 257: The reason of unavailability of IM BPG in Japan could be provided in short or with a cited reference.

Response: Thank you for your comment. We unfortunately could not find out the exact reason why IM BPG was withdrawn from the market in 1980s and had not been available until late 2021 in Japan. Some believe it is because a well-known person died of anaphylactic shock from IM BPG, but there are no detailed references.

8. Page 14, Line 264: The treatment efficacy of BPG enhanced with oral amoxicillin vs BPG alone was compared in a RCT conducted by Rolfs RT et al, which showed similar efficacy between the 2 treatment groups (83% vs 82%).

Response: Thank you for your comment. This sentence is not relevant, so we have corrected this sentence in order to avoid making unnecessary confusions (Line 275 in the clear version).

Tables:

9. Table 2: The coefficient and p values in multivariable analysis were different from that in the manuscript text.

Response: Thank you for your careful review. We corrected the manuscript (Lines 187-189 in the clear version).

10. Table 2: The reason that the variable "HIV-RNA <50 copies/mL" was not selected in the multivariable analysis should be stated.

Response: Thank you for your important point. In the univariable analysis, HIV-RNA <50 copies/ml was significantly associated with slower RPR decline after treatment. However, as clearly mentioned in the Methods (Lines 147-148 in the clear version), variable selection for the multivariable model was conducted by backward elimination using the P-values obtained with the Wald test. HIV-RNA <50 copies/mL became not significant in the multivariable model, therefore, it was dropped from the final multivariable model. In the previous studies, RPR responses were slower in HIV-untreated patients (who were supposedly with higher HIV-RNA). In our study, HIV-untreated patients were younger and had less history of syphilis, which were likely to confound the univariable results for HIV-RNA <50 copies/mL. We added a brief explanation to the results section (Lines 190-191 in the clear version).

The authors have no conflicting financial interests, all authors concur with the submission and have contributed to this work, and the material submitted has not been previously reported nor is under consideration for publication elsewhere. 

Sincerely,

Kazuaki Fukushima, M.D., Ph.D.

---

## [Decision Letter · Decision Letter 1]

21 Jun 2023

PONE-D-22-35264R1Changes in rapid plasma reagin titers in patients with syphilis before and after treatment: a retrospective cohort study in an HIV/AIDS referral hospital in TokyoPLOS ONE

Dear Dr. Fukushima,

Thank you for submitting your manuscript to PLOS ONE. After careful consideration, we feel that it has merit but does not fully meet PLOS ONE’s publication criteria as it currently stands. Therefore, we invite you to submit a revised version of the manuscript that addresses the points raised during the review process.

The revised version has been appreciated but I strongly sugggest to update the references' list and to improve the discussion

We look forward to receiving your revised manuscript.

Kind regards,

Antonella Marangoni, Ph.D.

Academic Editor

PLOS ONE

Additional Editor Comments:

The referee appreciated the revised version of the manuscript and I agree with the few comments.

In particular, you should update the bibliography as suggested and expand the discussion by including also other clinical conditions that might influence the serological response.

Reviewers' comments:

Reviewer's Responses to Questions

**Comments to the Author**

1. If the authors have adequately addressed your comments raised in a previous round of review and you feel that this manuscript is now acceptable for publication, you may indicate that here to bypass the “Comments to the Author” section, enter your conflict of interest statement in the “Confidential to Editor” section, and submit your "Accept" recommendation.

Reviewer #1: All comments have been addressed

2. Is the manuscript technically sound, and do the data support the conclusions?

Reviewer #1: Yes

3. Has the statistical analysis been performed appropriately and rigorously? 

Reviewer #1: Yes

4. Have the authors made all data underlying the findings in their manuscript fully available?

Reviewer #1: Yes

5. Is the manuscript presented in an intelligible fashion and written in standard English?

Reviewer #1: Yes

6. Review Comments to the Author

Reviewer #1: Reviewer's report

Title of the article

Changes in rapid plasma reagin titers in patients with syphilis before and after treatment: a retrospective cohort study in an HIV/AIDS referral hospital in Tokyo

Reference number

PONE-D-22-35264R1

Comments:

Thank you for the opportunity to review again your interesting manuscript. I enjoyed reading it as I feel the manuscript improved and suitable for publication. I recommend updating the bibliography (see Major Compulsory Revisions), and expanding the discussion in certain points (see Major Compulsory Revisions).

- Major Compulsory Revisions

Line 64-66: Please, try to update the bibliography as, for instance, reference [2] goes back to 1968. There are several most recent manuscripts that can be cited here [Tiecco G, Pathogens. 2021 or Ramchandani MS, Infect Dis Clin North Am. 2023].

Line 154-160: Please, add percentage in the main text in order to make data more easily comparable.

Line 232-236: Please, expand the discussion by including also other clinical conditions that might influence the serological response, and, if it is possible, compare your data to the available literature. Post-treatment RPR might be different in case of first infection treatment or syphilis reinfection management [Marchese V, J Clin Med. 2022]. Moreover, several atypical conditions might exist especially in PLWH such as serological-non-responder or serofast status [Seña AC, BMC Infect Dis. 2015 or Ghanem KG, Sex Transm Dis. 2021].

- Minor Essential Revisions

None

Level of interest

This is an interesting work as serological trend in syphilis infection is a tricky issue which requires further studies to better understand both the serological natural course and the serological response to treatment.

Quality of written English

English syntax is optimal.

Statistical review

This is a methodologically rigorous work.

7. PLOS authors have the option to publish the peer review history of their article (what does this mean?). If published, this will include your full peer review and any attached files.

Reviewer #1: No

---

## [Author Response · Author response to Decision Letter 1]

3 Jul 2023

Reviewer 1

1. Line 64-66: Please, try to update the bibliography as, for instance, reference [2] goes back to 1968. There are several most recent manuscripts that can be cited here [Tiecco G, Pathogens. 2021 or Ramchandani MS, Infect Dis Clin North Am. 2023].

Response: Thank you for your advice. We have added the reference you suggested (Line 66).

2. Line 154-160: Please, add percentage in the main text in order to make data more easily comparable.

Response: Thank you for your advice. We have added the percentage of included patients (Line 155-158).

3. Line 232-236: Please, expand the discussion by including also other clinical conditions that might influence the serological response, and, if it is possible, compare your data to the available literature. Post-treatment RPR might be different in case of first infection treatment or syphilis reinfection management [Marchese V, J Clin Med. 2022]. Moreover, several atypical conditions might exist especially in PLWH such as serological-non-responder or serofast status [Seña AC, BMC Infect Dis. 2015 or Ghanem KG, Sex Transm Dis. 2021].

Response: Thank you for your valuable comments. We have added an explanation addressing the potential impact of prior syphilis infection on post-treatment RPR change and the potential influence of serofast and serological non-responder, which are commonly observed in HIV patients. We have also added the references you suggested (Lines 242-246).

---

## [Editor Report · Decision Letter 2]

18 Jul 2023

PONE-D-22-35264R2Changes in rapid plasma reagin titers in patients with syphilis before and after treatment: a retrospective cohort study in an HIV/AIDS referral hospital in TokyoPLOS ONE

Dear Dr. Fukushima,

Thank you for submitting your manuscript to PLOS ONE. After careful consideration, we feel that it has merit but does not fully meet PLOS ONE’s publication criteria as it currently stands. Therefore, we invite you to submit a revised version of the manuscript that addresses the points raised during the review process.

The manuscript has definitely improved, but some further modifications are still necessary.

In particular, I think that the discussion should be improved. 

We look forward to receiving your revised manuscript.

Kind regards,

Antonella Marangoni, Ph.D.

Academic Editor

PLOS ONE

Additional Editor Comments:

Dear authors

The manuscript has definitely improved, but some further modifications are still necessary.

In particular, I think that the discussion should be improved. Moreover, please fulfill all the requests of the referee

---

## [Author Response · Author response to Decision Letter 2]

1 Aug 2023

Reviewer 1

Comment 3. Line 232-236: Please, expand the discussion by including also other clinical conditions that might influence the serological response, and, if it is possible, compare your data to the available literature. Post-treatment RPR might be different in case of first infection treatment or syphilis reinfection management [Marchese V, J Clin Med. 2022]. Moreover, several atypical conditions might exist especially in PLWH such as serological-non-responder or serofast status [Seña AC, BMC Infect Dis. 2015 or Ghanem KG, Sex Transm Dis. 2021].

Based on this comment, we revised the discussion section. In this latest revision, we have focused on the factors that influence the speed of RPR decrease after treatment and conducted a thorough comparison of our study with relevant references (Lines 234-255). Additionally, we have added a new paragraph about the complex effect of HIV on RPR dynamics based on provided references (Lines 256-265).

Thank you once again for your careful consideration of our work. We look forward to receiving further guidance from you regarding the next steps in the publication process.

The authors have no conflicting financial interests, all authors concur with the submission and have contributed to this work, and the material submitted has not been previously reported nor is under consideration for publication elsewhere. 

Sincerely,

Kazuaki Fukushima, M.D.

---

## [Decision Letter · Decision Letter 3]

12 Sep 2023

Changes in rapid plasma reagin titers in patients with syphilis before and after treatment: a retrospective cohort study in an HIV/AIDS referral hospital in Tokyo

PONE-D-22-35264R3

Dear Dr. Fukushima,

We’re pleased to inform you that your manuscript has been judged scientifically suitable for publication and will be formally accepted for publication once it meets all outstanding technical requirements.

Kind regards,

Antonella Marangoni, Ph.D.

Academic Editor

PLOS ONE

Additional Editor Comments (optional):

Reviewers' comments:

Reviewer's Responses to Questions

**Comments to the Author**

1. If the authors have adequately addressed your comments raised in a previous round of review and you feel that this manuscript is now acceptable for publication, you may indicate that here to bypass the “Comments to the Author” section, enter your conflict of interest statement in the “Confidential to Editor” section, and submit your "Accept" recommendation.

Reviewer #1: All comments have been addressed

Reviewer #2: All comments have been addressed

2. Is the manuscript technically sound, and do the data support the conclusions?

Reviewer #1: Yes

Reviewer #2: Yes

3. Has the statistical analysis been performed appropriately and rigorously? 

Reviewer #1: Yes

Reviewer #2: Yes

4. Have the authors made all data underlying the findings in their manuscript fully available?

Reviewer #1: Yes

Reviewer #2: Yes

5. Is the manuscript presented in an intelligible fashion and written in standard English?

Reviewer #1: Yes

Reviewer #2: Yes

6. Review Comments to the Author

Reviewer #1: Thank you for the opportunity to review again your interesting manuscript. I feel the manuscript improved and suitable for publication.

Reviewer #2: (No Response)

7. PLOS authors have the option to publish the peer review history of their article (what does this mean?). If published, this will include your full peer review and any attached files.

Reviewer #1: No

Reviewer #2: No

---

## [Editor Report · Acceptance letter]

20 Sep 2023

PONE-D-22-35264R3 

Changes in rapid plasma reagin titers in patients with syphilis before and after treatment: a retrospective cohort study in an HIV/AIDS referral hospital in Tokyo 

Dear Dr. Fukushima:

I'm pleased to inform you that your manuscript has been deemed suitable for publication in PLOS ONE. Congratulations! Your manuscript is now with our production department. 

Kind regards, 

on behalf of

PhD Antonella Marangoni 

Academic Editor

PLOS ONE